# Modulation of Secondary Cancer Risks from Radiation Exposure by Sex, Age and Gonadal Hormone Status: Progress, Opportunities and Challenges

**DOI:** 10.3390/jpm12050725

**Published:** 2022-04-30

**Authors:** Anat Biegon, Siobhan Cohen, Dinko Franceschi

**Affiliations:** Department Radiology, Stony Brook University School of Medicine, Stony Brook, NY 11794, USA; siobhan.cohen@stonybrook.edu (S.C.); Dinko.Franceschi@stonybrookmedicine.edu (D.F.)

**Keywords:** cancer therapy, ionizing radiation, sex differences, menstrual cycle, menopause, targeted radionuclide therapy, theranostics

## Abstract

Available data on cancer secondary to ionizing radiation consistently show an excess (2-fold amount) of radiation-attributable solid tumors in women relative to men. This excess risk varies by organ and age, with the largest sex differences (6- to more than 10-fold) found in female thyroid and breasts exposed between birth until menopause (~50 years old) relative to age-matched males. Studies in humans and animals also show large changes in cell proliferation rates, radiotracer accumulation and target density in female reproductive organs, breast, thyroid and brain in conjunction with physiological changes in gonadal hormones during the menstrual cycle, puberty, lactation and menopause. These sex differences and hormonal effects present challenges as well as opportunities to personalize radiation-based treatment and diagnostic paradigms so as to optimize the risk/benefit ratios in radiation-based cancer therapy and diagnosis. Specifically, Targeted Radionuclide Therapy (TRT) is a fast-expanding cancer treatment modality utilizing radiopharmaceuticals with high avidity to specific molecular tumor markers, many of which are influenced by sex and gonadal hormone status. However, past and present dosimetry studies of TRT agents do not stratify results by sex and hormonal environment. We conclude that cancer management using ionizing radiation should be personalized and informed by the patient sex, age and hormonal status.

## 1. Introduction

Exposure to external ionizing radiation in the context of war, nuclear accidents and medical procedures poses well-known health risks, including acute radiation sickness (tissue reaction or deterministic effects), which may result in death, as well as an increase in the risk of cancer in the years following exposure (stochastic effects) [1,2]. The risk of radiation-induced cancer increases with dose and decreases with age at exposure, and is further modulated by the organ/tissue involved and biological sex [1,2,3], such that the risks of radiation-induced solid tumors are consistently (approximately 2-fold) higher in women relative to men, regardless of the exposure mechanism. In particular, breast and thyroid in young girls and women exhibit the largest increase in relative and absolute risk of radiation-induced tumors [3,4,5,6,7]. Importantly, despite the well-known effects of female gonadal hormones, especially estradiol, on cell cycle, proliferation and differentiation in breasts and other estrogen-receptor-containing organs, including the thyroid [8,9,10], gonadal hormone status (e.g., puberty, stage of the menstrual cycle, menopause) at the time of exposure has not been included in most published studies on cancer risks related to exposure to external ionizing radiation [3,4,5,6,7].

Compared to external radiation risks, literature on sex differences in secondary cancer risks due to internal radiation exposure following ingestion of contaminated foods or diagnostic/therapeutic injection of radioactive compounds is sparser and sometimes contradictory [11,12,13,14,15,16,17,18], which is possibly related to the acknowledged uncertainties in estimating organ dose under these circumstances [11]. Despite these caveats, the majority of publications agree that the risk of secondary cancer among those exposed to internal radiation are similar to those reported in the literature on exogeneous radiation. In this regard, it is important to note the recent growth in the introduction and development of new injectable radiopharmaceuticals intended to accumulate preferentially in target tissues, namely targeted radionuclide therapy and theranostic agents [19,20]. As the number of patients exposed to such agents is growing exponentially, it is becoming increasingly relevant to consider the personalization of radiation doses administered in a medical context by sex and hormonal status to optimize the use of diagnostic and therapeutic radiation. Below, we highlight data supporting this notion, some putative mechanisms relevant to sex and gonadal hormone effects on radiation-related cancers and their potential implications for clinical practice.

## 2. Sex and Age Modulate Cancer Risks from Wartime Exposure to External Radiation

Following the atomic bombing of Hiroshima and Nagasaki in 1945, survivors have been surveilled for decades for cancer and other health consequences of the exposure to radiation. Noting the location of the survivors relative to the epicenter, the radiation dose absorbed by individuals could be calculated. This information helped create the largest extant database on health risks from exposure to ionizing radiation, describing the rates of radiation-attributable cancers in males and females as a function of age at exposure [3,21,22,23,24]. While all publications report a higher rate of solid tumors in women relative to men, regardless of the model (e.g., no dose effect, linear dose–response, a linear-quadratic dose–response model), the NRC publication [3] has the most complete breakdown of attributable risk by sex, organ and age at exposure and is therefore the most amenable to exploring sex differences and, using age as a surrogate of puberty and menopause, exploring possible dependence on gonadal hormones. In Table 1, we have used these data to highlight the relationship between sex and age before puberty (0), after puberty but before menopause (age 15) and after menopause (age 60) and the risk of various cancers.

In Figure 1, we have provided a graphical representation of the dependence of various cancers on sex and age at exposure.

Perusal of Table 1 and Figure 1 demonstrates the expected negative correlation between age at exposure and rates of radiation-attributable cancer in both sexes, but also highlights remarkable sex differences in the relationship between risk and age at exposure. In aggregate, the lifetime attributable risk of all cancers is substantially higher for women exposed at a younger age (under 50, the average age of menopause in Japan, [25]) compared to men exposed in the same age range, with a female/male ratio of ~2 for all solid tumors (Table 1). Importantly, 50 is the average age of menopause onset, so the disappearance of the sex difference around this age suggests modulation of risk by female gonadal hormone status. The increased risk in women is greatly attenuated with age, such that the rate of radiation-attributable solid tumors in aggregate is only 20% higher in women exposed at age 60 relative to men exposed at the same age. 

As can also be seen in Table 1 and Figure 1, the relative risk of radiation-induced cancer in men and women exposed at different ages varies among different organs. The organs most susceptible to radiation-induced cancer in those exposed to external radiation at a young age is the female breast, followed by the female lung and the female thyroid. Furthermore, the relationship between radiation-attributable breast and thyroid cancer and age at exposure is sexually dimorphic, with considerably steeper initial slopes in females compared to males (Figure 1 top row). To illustrate, risks of breast and thyroid cancers attributable to external radiation are highest in females exposed at age 0. The attributable rate of secondary breast cancer in females is 12.6-fold the attributable risk of prostate cancer in males exposed at the same age, and the ratio for thyroid cancer is 5.5. The vulnerability of girls relative to boys remains high, though decreasing, through puberty (age 15). After age 50, when most women reach menopause, the relative risk is decreased to the point where there is no longer a meaningful difference between men and women, indicating some modulation by hormonal status (Table 1, Figure 1). 

This pattern appears to be unique to breast and thyroid, and likely explains the female-specific shift in the likelihood (rank order) of various cancers as a function of age/hormonal status. 

Although female radiation-attributable lung cancer rates are higher than those of males, the ratio of female to male remains constant and independent of age at exposure (2.33 at age 0, 2.32 at age 15, 2.48 at age 60). This sex difference disappears only in extreme old age, suggesting that, unlike breast and thyroid, the risk of radiation-induced lung cancer is modulated by age and sex, but not by hormonal status. Notably, the sex difference in the risk of lung cancer cannot be fully explained by differential effects of smoking, since the gender-averaged excess relative risk per Gy of lung cancer (at age 70 after radiation exposure at 30) was estimated as 0.59 for non-smokers, with a female–male ratio of 3.1 [26]. A smaller sex difference with a similar age-related pattern is observed with stomach cancer, where the female/male ratio is 1.65, 1.5 and 1.35 for exposure ages of 0, 15 and 60, respectively (Table 1).

Another observation suggesting a relationship between radiation-induced cancer and female gonadal status can be found when examining the relative susceptibility to various cancers as a function of age. Thus, the four most susceptible organs in young females (<50 years old at exposure) are (in descending order) breast, lung, thyroid and colon, while the rank order in older females is lung, bladder, colon and blood (leukemia), much more similar to the rank order in males (colon, lung, leukemia and bladder), which does not change appreciably with age at exposure between birth and old age (Table 1). A direct link with gonadal hormone staus was established in one of the most recent papers published on atomic bomb survivors, showing that a young age at menarche significantly increased radiation-attributable breast cancer risk [27].

Importantly, not all tumors are more susceptible to radiation in females: The risk of radiation-induced bladder cancer is identical in males and females regardless of the age at exposure (Figure 1, 2nd row, on the right), while the attributable risk of colon cancer, liver cancer and leukemia is higher in men, with a male to female ratio for exposure at birth 1.53, 2.18 and 1.28, respectively. The male-to-female relative risk remains fairly constant from exposure at age 0 to age 60 (±0.01 in colon, ±0.16 in leukemia, ±0.25 in liver, Table 1 and Figure 1, bottom). Thus, there is no significant change in male/female ratio of colon, liver or blood cancer around the onset of puberty or menopause, suggesting that radiation-associated cancer risk in these organs is not appreciably modulated by female or male gonadal hormone status. 

## 3. Sex and Age Modulate Secondary Cancers Attributable to External Radiation Exposure in Medical Practice 

### 3.1. Radiation Therapy

The best source of information on the effects of therapeutic radiation on secondary cancer risk derive from survivors of Hodgkin’s Lymphoma (HL) treated with radiation, who have been observed in a number of studies designed to monitor the rate of secondary cancers in this population over an extended time period [28,29,30,31,32,33,34,35,36,37,38]. While much smaller in scope than the atomic bomb survivor studies, these studies consistently found an increased rate of solid tumors in HL patients treated with external radiation [28,29,30,31,32,33,34,35,36,37,38], although, like in atomic bomb survivors (above), the risks were modulated by age at exposure, sex and site of cancer. Thus, HL patients treated with external radiation have a higher incidence of all solid cancers except for prostate [28], and the largest increase was in (female) breast cancer. The absolute excess risk for breast cancer following external radiation treatment was between 57 and 121.8 per 10,000 person years [28,36]. Importantly, the risk was dependent on radiation volumes, with the highest cancer risk associated with mantle field radiation, which broadly irradiates all of the lymph node areas in the upper half of the body [36], as well as by hormonal status, such that early menopause significantly reduced the risk of secondary breast cancer [37].

Notably, radiation treatment of childhood cancer appears to be associated with significant increases in breast cancer risk, comparable to those reported in females with a BRCA gene mutation [39,40]: both BRCA and certain radiation treatments carry similarly high risks compared to the cumulative population incidence of breast cancers diagnosed before age 45, namely 12–16% in patients treated with radiation as children, 13–20% in women with a BRCA mutation, and 1% in the general population [28,29,30,31,32,33,34,35,36,37,38,39,40]. The large size of the increased risk of breast cancer in females exposed to radiation in childhood led to the development of specific recommendations for breast cancer surveillance for female survivors of childhood, adolescent, and young adult cancer given chest radiation [41], as recommended for women harboring harmful BRCA mutations [42]. Notably, unlike the genetic risk, breast cancer risk due to exposure to radiation in childhood might be reduced by personalizing radiation treatment parameters considering sex and hormonal status. 

The higher vulnerability of females to radiation-induced thyroid cancer, noted above for atomic bomb survivors, is also seen in the medical context. In a relatively large cohort of subjects given radiation therapy for a variety of childhood cancers, including HL, leukemia, central nervous system cancer and soft tissue sarcoma [43], relative risk of a secondary thyroid cancer was increased 2.3-fold in females relative to males. This study also demonstrated the expected inverse relationship between age at exposure and risk of secondary cancer. 

A more recent analyses of this population [44,45] confirmed the findings of increased risk of breast, thyroid and other cancers in the aftermath of childhood cancer radiation therapy, and reiterated the observation supporting a relationship between female gonadal hormone status (i.e., menopause) and the risk of radiation-induced breast cancer. Thus, the excess risk of radiation-related breast cancer was sharply reduced among women who received 5 Gy or more to the ovaries—a dose likely to disrupt ovarian function [44]. The authors concluded that “Irradiation of the ovaries at doses greater than 5 Gy seems to lessen the carcinogenic effects of breast irradiation, most likely by reducing exposure of radiation-damaged breast cells to stimulating effects of ovarian hormones” [45].

Regarding treatment of non-pediatric patients, studies of secondary cancers associated with radiation treatment of breast cancer consistently reveal a statistically significant increase in the risk of subsequent cancer in the contralateral breast [46]. Which was modulated by dose and age at exposure, such that women < 40 years of age who received >1.0 Gy of absorbed dose to the contralateral breast had a 2.5-fold greater risk for contralateral breast cancer than unexposed women (RR = 2.5, 95% CI 1.4–4.5).

Radiation treatment of breast cancer is also associated statistically significant increase in the risk of lung and other cancers [47,48,49,50]. Individual cohort studies as well as a metanalysis of 22 studies encompassing 245,575 irradiated and 277,164 non-irradiated women show that the rate of secondary primary lung cancer increases (8.5% per Gray] with dose [47] and time, peaking 15 years after breast cancer irradiation [48]. Summary values for risk (expressed as standardized incidence ratio) in this time window were 1.91 for lung, 2.71 for esophagus and 3.15 for thyroid. Other cancers showing significant elevation in risk included contralateral breast and sarcomas [47,48,49,50]. Notably, incidence of secondary lung cancer in women increased after radiation therapy but decreased after chemotherapy for breast cancer [50]. The authors note that although the absolute risk is relatively low, the growing number of long-time survivors after breast cancer treatment highlights the need for advances in normal tissue sparing radiation techniques [47,48]. Encouragingly, some novel approaches recently described appear to achieve this end, based on dosimetric calculations [51]. 

Finally, the use of external ionizing radiation for therapy is not limited to cancer [4,52,53,54,55]. In this regard, it is noteworthy that extended follow-up of individuals administered X-ray therapy to treat tinea capitis in children revealed a significant increase in risk of thyroid cancer, with the excess risk being up to 4-fold higher in females relative to males [4,53]. Similar results were reported by Shore et al. [54] from a study of thyroid tumors following thymus irradiation. The authors report that for both thyroid cancers and adenomas the absolute excess risk per rad was two to three times as great in females relative to males. An increased risk of breast cancer was noted as well [52,55].

### 3.2. Sex and Age Modulate Secondary Cancers Attributable to Radiation from Medical Imaging

Diagnostic procedures using radiation result in organ exposures which are considerably lower than those employed in radiation therapy, although cumulative effects may result in a substantial radiation dose [56]. Furthermore, accumulating evidence suggests that such diagnostic imaging procedures may result in a significant increase in the risk of secondary cancer [6,7,57,58,59,60,61,62,63]. These observations led to the 2019 publication of a white paper emphasizing the need to reduce unnecessary radiation exposure from medical imaging [64]. Importantly, examination of the results of specific studies reveals patterns of organ, sex and age vulnerability similar to those described above for wartime and therapeutic radiation exposures. Thus, a 2000 study of the U.S. scoliosis cohort [57,58] reported 77 breast cancer deaths compared with the 45.6 deaths expected on the basis of U.S. mortality rates (standardized mortality ratio [SMR] = 1.69; 95% confidence interval [CI] = 1.3–2.1). Furthermore, risk was found to increase significantly with an increasing number of scans and with cumulative radiation dose. Notably, all of the women in the cohort were diagnosed before age 20 (mean age 10.6 years), and the average length of follow-up was ~40 years. Other cancer deaths (e.g., liver) were not significantly increased [58,59]. A 2009 study of cancer risk related to CT angiography [59] estimated that 1 in 270 women who undergo CT coronary angiography at the age of 40 years will develop cancer from that scan, compared with 1 in 600 men (more than 2-fold female/male risk ratio). A subsequent study of low-dose ionizing radiation of patients following a myocardial infarction [61] similarly showed that exposure to radiation from cardiac imaging was associated with an increased risk of cancer. When stratified by sex, the adjusted HR for incident cancer was higher among women relative to men exposed to low-dose ionizing radiation (HR 1.005, 95% CI 1.003–1.007 in women, and 1.002, (95% CI 1.001–1.003 in men) and the interaction term for sex and radiation exposure in a multivariable model was statistically significant (*p* < 0.001). Notably, the majority of patients in this study were elderly males, and the menopausal status of the women or the specific organ/site of the cancer (e.g., thoracic, including breast, rather than breast, lung, etc.) were not mentioned. Comparing radiation dose and cancer risk associated with whole body PET/CT scanning in the U.S. and Hong Kong, Huang et al. [6] showed that for 20-year-old U.S. women, lifetime attributable risks (LARs) of cancer incidence were between 0.231% and 0.514%, and for 20-year-old U.S. men, LARs of cancer incidence were lower (between 0.163% and 0.323%). The risks were higher for the Hong Kong population overall but the excess risk in women relative to men was the same. When Rampinelli et al. [7] investigated the effect of low-dose CT for lung cancer screening on the risk of cancer, they, too, found LARs in women to be 3- to 4-fold higher than for men. These findings were confirmed most recently in a paper comparing risk/benefit ratios of different screening scenarios [61], showing that for an annual low-dose CT screening of (ex-)smokers aged between 50 and 75 years the estimated radiation-related lifetime attributable risk to develop was moree than doubled in women compared to men, resulting in a significantly lower benefit–risk ratio in women (10) relative to men (25). 

Findings in children also reiterate the higher vulnerability of young girls to diagnostic radiation-induced cancer relative to boys of the same age, noted in the atomic bomb survivor cohort. In 2012 and 2013, two groups published studies on the effects of CT scanning in children on cancer risk [62,63]. The first study focused on leukemia and brain tumors and showed a higher excess relative risk in girls [0.026 ERR/mGy vs. 0.016] although the difference was not statistically significant [*p* = 0.08] [63]. The latter, larger study demonstrated a large sex difference in the risk of solid tumors in children exposed before the age of 5, with girls at a 2- to 6-fold higher risk relative to boys, depending on the type of scan (head, pelvis, spine, etc.) [63]. 

These and additional observations led to the publication of a white paper by the FDA, calling for the reduction of unnecessary radiation exposure from medical imaging [64]. In this paper, authors suggest that “In some cases, ordering physicians may lack or be unaware of recommended criteria to guide their decisions about whether or not a particular imaging procedure is medically efficacious. As a result, they may order imaging procedures without sufficient justification and unnecessarily expose patients to radiation. Various professional societies and organization have developed and are working to disseminate imaging referral criteria, called “appropriateness criteria” or “appropriate use criteria,” associated with a number of medical conditions. However, criteria for appropriate ordering of medical imaging exams have not yet been broadly adopted by the practicing medical community.

For this situation to be ameliorated, it may also be necessary to increase awareness and acceptance of the risks associated with medical imaging, which are substantially more controversial than those associated with atomic bomb or radiation therapy, with studies more likely to be criticized for using inappropriate models or control groups [65]. 

In this regard, the very large sex differences in effect size and the age trajectory summarized above and lead to the interesting possibility that some of the variability in the results of studies on the delayed cancer risks associated with low-dose radiation [55] derive from a failure to stratify patients by sex and age/hormonal status. 

This is an ongoing problem. Thus, a very recent study showing that low-dose radiation exposure from nuclear medicine examinations (excluding PET/CT) is significantly associated with a higher risk of cancer and benign central nervous tumors in pediatric patients [66], leading the authors to recommend that “nuclear medicine practitioners, including physicians and technologists, should show more initiative to protect patients from radiation and should be prudent in optimizing the radiation dose in pediatric nuclear medicine procedures”. Unfortunately, the results of this study were not stratified by sex.

## 4. Effects of Accidental Exposure to Internal Radiation

The nuclear accident at Chernobyl in 1986 exposed people in the region, mostly to internal radiation through ingestion of contaminated foods. Early reports indicated that in the next few following years, the incidence of thyroid cancers in the exposed population increased [12,13,14,15]. As expected, iodine radioisotopes from the fallout concentrated in the thyroid, which is particularly sensitive in children due to rapid cell proliferation [13]. In addition to young age, female sex was also mentioned as an important risk factor, but while the studies on cancer incidence following Chernobyl agree that there is a significant negative correlation between age of exposure and elevated radiation-attributable thyroid cancer incidence [12,13,14,15], sex differences and hormonal effects were not as clear-cut. While thyroid cancer rates were higher in girls than in boys exposed to the same dose of radiation in the Chernobyl studies [13], the sex difference appeared to be smaller than those reported from unexposed populations elsewhere [67]. This uncertainty echoes the CERRIE report [11] but may also be more specifically related to the specific incident.

This ambiguity could be due to the circumstances surrounding the nuclear accident at Chernobyl, the data may not be entirely accurate. Determination of internal radiation dose following the accident was not definitive due to unknown thyroid mass of individuals and variability in diet [12]. Data was not collected systematically, as in the case of Hiroshima and Nagasaki, rather, analysis has been performed on data collected through hospital records and regional thyroid cancer registries [12,13,14,15].

Radiation exposure due to inhalation of radioactive isotopes that concentrated in the lungs of people in close proximity to the accident may have created unstudied elevated cancer risks in nearby tissues such as the heart and breast [15]. High iodine uptake of organs beside the thyroid may also elevate cancer risk, since iodine uptake in healthy breast is slightly elevated, and significantly elevated in lactating females [16]. Notably, more recent reports on Chernobyl, representing longer follow-up periods, do indeed show elevated risk of breast cancer [68], especially in women who were lactating [Cahoon1], who demonstrated a greater than 2-fold increase in the risk of breast cancer compared with the general population (SIR = 2.49 (95% CI: 1.55, 3.75) [69]. Recent studies also confirm the expected sex difference in risk thyroid cancers and nodules, which were singifciantly increased in females [70]. 

The Techa River incident (1956) is another case involving a moderately-sized cohort, with the most recent new smoking-adjusted and site-specific cancer risk estimates demonstrating a statistically significant (*p =* 0.02) linear trend in all-solid cancer incidence risks. Examination of site-specific risks revealed statistically significant radiation dose effects only for cancers of the esophagus and uterus with an ERR per 100 mGy estimates in excess of 0.10. Esophageal cancer risk estimates were modified by ethnicity and sex but not smoking [71].

Subjects exposed in the most recent nuclear accident (Fukushima 2011) are being actively followed and it may be too early to observe effects on breast cancer and other slow-developing tumors. Available reports on thyroid cancer did demonstrate increased rate of thyroid cancer detection in this population, but it was suggested these are not due to radiation exposure but rather to the intensive screening methods used [72]. 

## 5. Sex, Hormonal Status and Targeted Radionuclide Therapy: A Challenge and an Opportunity 

Recent advances in cancer therapy with radioactive substances involve parenteral administration of a radiopharmaceutical or radiolabeled antibody which is designed to preferentially accumulate in neoplasms, termed Targeted Radionuclide Therapy (TRT) and Radioimmunotherapy, respectively [19,20,73,74]. Development of new TRT agents is increasingly accompanied by the use of companion diagnostics and individual dosimetry [75,76] since, as stated in Roncali et al. [61], systemic radiopharmaceutical therapy with associated companion diagnostics “is the embodiment of precision medicine” and dosimetry is the optimal tool to predict individual safety and efficacy of TRT [75].

### 5.1. Sex and Age Modulate Cancer Risks from Radioactive Iodine Therapy of Thyroid Disease

The earliest example of TRT is the use of radioactive iodine for treatment of thyroid cancer [77,78]. An early study of I-131 therapy focusing on female reproductive function reported slightly earlier onset of menopause in some women treated with I-131 and some occurrences of spontaneous and induced abortion in the first 12 months following treatment, but no significant decrease in fertility and no increase in birth defects [79]. The same study reported an increased risk of secondary breast cancer, further supported by a more recent study demonstrating a significant association between radioiodine dose and risk of secondary breast cancer [80]. 

Hyperthyroidism is also treated with radioactive iodine, though at lower doses than those used for treating thyroid cancer. A 2019 study of 18,805 patients showed that in radioactive iodine-treated patients with hyperthyroidism, greater organ-absorbed doses were modestly though statistically significantly associated with an increased risk of female breast cancer [17]. The results of this study were criticized by others [18], though the relationship between dose and cancer risk was later supported in a very recent meta-analysis of cancer risks following radioactive iodine therapy for hyperthyroidism [5], which reported a significant association between radioactive iodine dose and breast cancer mortality. The study also reported a significant elevation in thyroid cancer incidence and mortality which was not stratified by sex—rather, they compared risks in studies including more than 80% females to those including 80% or less [5], probably since females represented the large majority of patients in all studies [17].

The strong relationship between sex, age, hormonal status and potential risks of radioactive iodine was recently highlighted in a 2021 paper by Al-Jabri and colleagues [81]. The stated aim of the study was to investigate whether Tc-99 thyroid uptake may be used in place of I-131 uptake for implementing personalized treatments. The investigators found that patient age and the time difference between the two respective uptake measurements significantly influenced the uptake correlation in females but not in males, and uptake was correlated with hormone levels. Consequently, the uptake of the two tracers was significantly more correlated in males than in females (r^2^ = 0.71 vs. 0.38, *p* < 0.001) and estimating I-131 uptake based on Tc-99m uptake was predictive in male but not female patients (91% vs. 16%). The authors conclude that their finding highlight “the potential need for gender consideration when planning ^131^I patient management and when reporting studies results” [82]. This is in variance with the *American Thyroid Association’s Management Guidelines for Patients with Thyroid Nodules and Differentiated Thyroid Cancer* published in 2015 [81], which does not address sex by age interaction or hormonal status.

### 5.2. TRT for Central Nervous System, Neuroendocrine and Other Tumors: Too Early to Call?

In recent years, TRT has expanded to treat a wider range of targets and tumors in addition to thyroid, with a steadily increasing number of radiopharmaceuticals in preclinical and clinical development [19,20,73,74,83,84,85,86,87,88,89,90,91,92,93]. Consequently, TRT represents an unprecedented opportunity for personalizing medical radiation by sex and hormonal status before rather than after the patients who are the most vulnerable (namely, young females) and can benefit most from individualized, well-informed therapy, become statistics in the large and long follow-up studies necessary to establish secondary cancer risks from medicinal irradiation. 

To elaborate, clinical trials using I-131 MIBG, e.g., [86,87,88], did include females, but did not analyze any risks by sex and age/hormonal status. Since neuroblastoma is typically treated during childhood and is associated with increased risk of secondary cancer [94,95], it is particularly important to understand the specific long-term impact of radionuclide therapies on young females. In this regard, while gonadal failure following I-131 MIBG treatment had previously been attributed to damage caused by concurrent chemotherapy, the first two cases of ovarian failure in patients treated with only I-131 MIBG for childhood neuroblastoma were reported in 2014 [95], suggesting that it was the radiation treatment which damaged the female gonads. It is noteworthy in this regard that MIBG targets the norepinephrine transporter, the function of which has been shown to be sex dependent and modulated by the menstrual cycle [96,97]. 

Similarly, various TRT radiopharmaceuticals targeting somatostatin-receptor (SSTR) positive neoplasms are increasingly in use in trials and practice to treat neuroendocrine tumors, which are also relatively common in children, but long-term follow-up and risks of secondary cancer which take a long time to develop, such as breast cancer, have not been published to date; and studies of acute and subacute toxicity do not stratify results by sex, age or hormonal status [98]. Notably, SSTR are expressed in the human endometrium and appear to change locations throughout the menstrual cycle [99,100]. Furthermore, estradiol has been shown to affect somatostatin receptor expression in female rat pituitary and human prostate cells in culture [101,102]. 

Inflammatory markers (beta-3-integrin, interleukin-6, PGE2 receptor types EP2/EP4, and COX-1) which are overexpressed in tumor vasculature also serve as potential targets for radionuclide therapy [84,103,104]. All of these targets are expressed in the female reproductive system and their expression varies significantly across the menstrual cycle [105,106,107]. Yet another example is the Orexin receptor type 1 (OX1R), a GPCR cell surface receptor expressed in peripheral cancers and a putative target for TRT [91]. The expression of this receptor in animal models was shown to be upregulated in ovaries and modulated by the estrous cycle [108].

### 5.3. The Challenge of TRT Risk Prediction and Modulation in Females

TRT presents a challenge for risk modulation to healthy organs since healthy tissue sparing strategies which can be deployed with external radiation [51] are not relevant when the radioactive agent is administered systemically. Instead, TRT trials incorporate individual dosimetry to predict risk of toxicity as well as efficacy [98,109,110,111]. TRT clinical development may also involve the use of “theranostic” (or theragnostic) agents whereby a pair of radiopharmaceuticals targeting the same molecule are used to assess bio-distribution and “personalized” dosimetry with a low energy, short half-life isotope and achieve tumor response with a higher-energy emitter. Although the principle is not new, it is receiving a lot of attention recently [112,113,114,115,116,117]. This approach is predicated on the assumption that the bio-distribution of the target is stable within individuals over time, which is patently untrue for reproductively competent women who experience large changes in levels of multiple biomarkers during puberty, menstrual cycle, pregnancy, lactation and menopause [96,97,118,119]. These changes are likely to modulate the susceptibility of females to environmental hazards and influence the uptake of targeted therapeutic and diagnostic radiopharmaceuticals, their safety and efficacy [120,121,122,123,124,125,126,127,128,129,130,131,132,133]. 

Despite the solid evidence for sex and hormone modulation of current and potential radiopharmaceutical target abundance summarized above, dosimetry studies which address or stratify data by sex or hormonal status are extremely rare. Thus, a study of [18F]fluoroestradiol radiation dosimetry which included 49 women and 2 men mentions that 19 of the women were premenopausal, rather than just noting the age distribution. The authors do mention that the results of the two men were similar to those of the women, but there is no mention of stratification by hormonal status, menstrual cycle phase in the premenopausal women or direct measurement of uptake and organ dose in the ovary [131]. A subsequent dosimetry of another estrogen receptor targeting radiopharmaceutical (4-fluoro-11beta-methoxy-16alpha-18F-fluoroestradiol) ascertained hormonal status in all 10 participants (all women) [132]. The six premenopausal women were scanned during the early follicular stage of their menstrual cycle, when endogenous estrogen levels are at their lowest [133] and least likely to compete with the tracer for the receptor, thus providing the “worst case scenario” for organ exposure for this target. The authors reported significantly lower uterine uptake (%ID) at 120 min after injection in pre- relative to postmenopausal subjects (0.075 ± 0.033%ID and 0.163 ± 0.026%ID, respectively; *p* = 0.003, two-tailed unpaired *t*-test). The only study reporting on the effect of menstrual cycle phase and menopause on tracer uptake and organ dose was published in 2015 [126]. In this study, ^11^C-vorozole, an aromatase inhibitor, was injected in 13 men and 20 women (10 premenopausal and 10 post-menopausal) and the young (premenopausal) women were scanned at 2 discrete phases of the menstrual cycle (midcycle and late luteal). The results were quite striking: Standardized uptake values and organ doses for (dominant) ovary in women scanned at midcycle were 3-fold and >30-fold higher, respectively, compared to the same women at other stages of the cycle as well as postmenopausal women, making the ovary (84.2 μSV/MBq), rather than liver (15.5), kidney (6.06) or spleen (6.1), the dose limiting organ for young ovulating women (Figure 2) [126]. Notwithstanding these findings, dosimetry studies for new TRT agents in development published 2 years later [134,135,136,137] do not stratify data by sex, menopause or menstrual cycle phase, nor do they even mention the sex of the subjects [137].

In addition to radionuclide accumulation, sex and hormonal status influence the pharmacokinetics and pharmacodynamics of TRT drugs. For example, biological sex and hormonal status affect distribution volume, protein binding, transport and metabolism of many classes of drugs [138,139]. Overall, women have lower BMI, have less surface area and less water in the body. Cardiac output and regional blood flow, particularly relevant to TRT because it is administered intravenously, varies between sexes [128]. Plasma protein binding is also sex-dependent: females taking estrogen-containing oral contraceptives have even more pronounced differences relative to males [139].

While correction for height, weight, surface area and body composition eliminate sex differences for some drugs, several drugs still affect women and men differently due to large differences in metabolism and elimination. Hepatic metabolism of drugs by cytochrome p450-linked liver monooxygenases (CYPs) is modulated by gonadal hormones; men typically have a faster oxidative metabolism [138]. Renal elimination, measured by renal blood flow and GFR, is highest in pregnant women, followed by men and then non-pregnant women, while pulmonary elimination is highest in men, followed by pregnant women and then non-pregnant women [139]. Lastly, pharmacodynamics, including number of receptors, binding affinity and signal transduction, also vary by sex and hormonal status [99,100,105,106,107,108,127,139]. 

## 6. Summary and Path Forward

Therapeutic and diagnostic use of ionizing radiation in cancer and other indications is wide spread and increasing. Perusal of published reports in the last few decades demonstrate that:Solid tumor incidence secondary to exposure following the atomic bombing of Hiroshima and Nagasaki, the largest cohort subjected to continued surveillance over more than 60 years, is markedly higher (~2-fold) in women.This excess risk varies by organ and age at exposure, with the largest sex differences (6- to 10-fold) found in female thyroid and breasts exposed between birth and menopause (~50 years old) when compared to males in the same age group.The risk of secondary breast or thyroid cancer in females decreases more steeply with age at exposure in females compared to males.Secondary breast cancers take a long time (10–20 years) to manifest and the increased risk due to radiation is sustained for decades.The pattern of vulnerability is remarkably consistent across a variety of exposure mechanisms, radiation doses, mathematical models and risk metrics, such that young females are at a highly elevated risk of secondary solid tumors, especially breast and thyroid, when exposed to wartime, therapeutic or diagnostic radiation.Smaller cohort studies in humans and animals also show large changes in cell proliferation rates, radiotracer accumulation and target density in female reproductive and non-reproductive organs, including breast, thyroid and brain, in conjunction with physiological changes in gonadal hormone (especially estradiol) levels such as occur during the menstrual cycle and menopause. NB, while these effects may explain higher propensity of females to radiation induced cancers of the breast and the thyroid, they do not explain the sex difference in lung cancer, the biological origin of whch is still unknown.

These sex differences and hormonal effects present challenges as well as opportunities to personalize radiation-based treatment and diagnostic paradigms so as to optimize the risk/benefit ratios in radiation-based cancer therapy and diagnosis. Specifically, Targeted Radionuclide Therapy (TRT) is a fast-expanding cancer treatment modality utilizing radiopharmaceuticals with high avidity to specific molecular tumor markers, many of which are influenced by sex and gonadal hormone status. However, past and present dosimetry studies of TRT agents do not usually tratify results by sex and hormonal environment, although it appears that the accuracy of cancer diagnosis and the safety and efficacy of cancer management using ionizing radiation may be improved if personalized and informed by the patient sex, age and hormonal status. Importantly, race/ethnicity and smoking do not appear to interact with or modulate the size and direction of the sex and hormone effects on radiation-induced cancer [25,140].

Since 1993, the FDA has required the inclusion of women in clinical trials and further advised separate analysis of male and female participants in clinical drug trials [141]. The failure to adopt the latter directive or to include information on female hormonal status in medical research is widespread and not unique to radiology [142,143,144,145]. Unfortunately, this omission results in a paucity of evidence-based assessment of secondary cancer risk from medical radiation exposure along the female life cycle [130,144]. This is probably the reason why guidelines and recommendations pertaining to the use of ionizing radiation in research and clinical practice [146,147,148,149,150,151], while invoking the need to take sex and biology into account, include specific recommendations to protect fetuses, infants and children but do not differentiate between young males and young (non-pregnant or lactating) females.

In this regard, the International Commission on Radiological Protection recommendations [146] are a case in point:

“ICRP has decided not to set sex-specific dose limits despite the fact that the carcinogenic risk (cancer incidence) for women is by a factor of 1.7 higher than for males (8, our reference [3]). Sex-specific dose limits would make regulations for radiological protection more complicated and could lead to discrimination at workplaces. ICRP has concluded that the dose limits have been set in such a dose range that women are also well protected. Therefore, the wT values have been sex-averaged…risk evaluations are needed. Then, individual parameters (e.g., sex, age and possibly other individual parameters) have to be used. Organ doses have to be assessed. Further, the individual biological variability has to be taken into account where such data are available. The wT for breast is increased, which is especially due to the higher breast cancer risk in juveniles and young women.” The most recent (2021) update to these guidelines reiterates that “it should also be recognised that lifetime cancer risks vary with age at exposure, sex and population group [147].

Similarly, the American College of Radiology practice parameters for nuclear medicine procedures only single out women in the context of pregnancy and lactation, where the impetus is protecting the fetus or newborn rather than the mother: “Administration of iodine-131 sodium iodide to pregnant or lactating patients (whether currently breastfeeding or not) is contraindicated. Complete cessation of breastfeeding 6 weeks prior to administration of iodine-131 sodium iodide is recommended to decrease the radiation absorbed dose to the maternal breast tissue and prevent the ingestion of radioactive breast milk by the nursing child” [150].

Another case in point is the guidelines for human dosimetry of ^11^C radiopharmaceuticals proposed by Zanotti-Fregonara and Innis in 2012 [152]. The authors concluded that 370 MBq of any ^11^C radiopharmaceutical could be safely used for first in-human studies without exceeding the upper limit of effective dose for research subjects set by regulatory agencies. However, the 2015 dosimetry study of N-Methyl-[^11^C]Vorozole, which reported direct measurements of ovarian exposure at different phases of the menstrual cycle [126] found the dose to the ovary at midcycle was 84.2 μSv/MBq, or 31.1 mSv/370 MBq, which would have exceeded the 30 mSv FDA limit for ovary exposure in research studies [153]. The maximal dose administered in the study was 296 MBq [126]; it is clear from the dosimetry results that higher doses could be safely administered to young women in the early follicular or late luteal phase of the cycle, as well as to men of all ages and postmenopausal women. The implication is that some sparing of non-target organs (especially reproductive organs, breast and thyroid) and consequently improvement of the risk/benefit ratio can be achieved if treatments are administered to young females during the least vulnerable period of the cycle, which varies among different radiopharmaceutical and targets [118,119,120,121,122,123,124,125,126,127]. Conversely, dosimetry studies should include the most vulnerable period [126,132]. As for the prepubertal population, where excess risks to female infants and young girls are the highest, it may be advisable to reconsider alternative diagnostic and therapeutic options if available. 

## Figures and Tables

**Figure 1 jpm-12-00725-f001:**
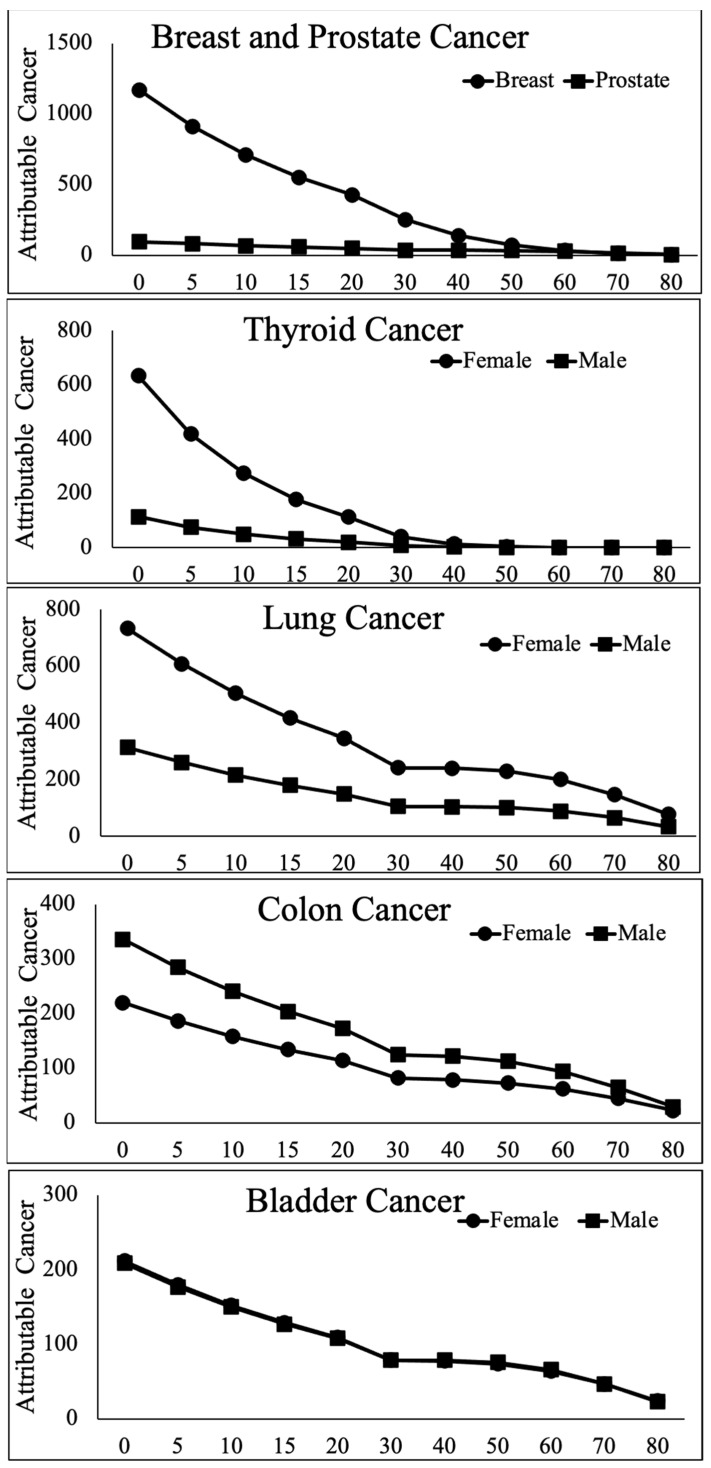
Cancers attributable to radiation exposure vary by organ, sex and age at exposure. Y axis: attributable cancer cases/100,000 persons/0.1 Gy. X axis: Age at exposure in years. Derived from [3].

**Figure 2 jpm-12-00725-f002:**
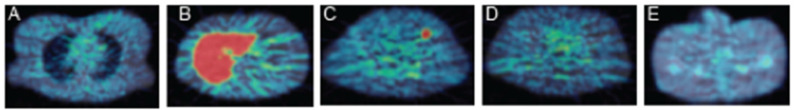
Effects of menstrual cycle on ovarian accumulation of [^11^C]vorozole, an aromatase inhibitor. Pseudocolored (rainbow spectrum) examples of ^11^C-vorozole uptake. axial images (**A**–**E**) were overlaid on the attenuation scan obtained immediately before the emission scan: (**A**) female, level of breasts and lungs; (**B**) female, level of liver; (**C**) female at midcycle, level of ovary; (**D**) female, level of lower pelvis; (**E**) male, level of lower pelvis. Quantitative analysis published in [126].

**Table 1 jpm-12-00725-t001:** Ranking Radiation-Attributable Cancer by Age, Sex and Organ (derived from NCR, [3]).

Exposure Age (Hormonal Stage)	Rank	Females Organ	Cases/100 K	Rank	Males Organ	Cases/100 K
Birth (0)	1	Breast	1171	1	Colon	336
Pre	2	Lung	733	2	Lung	314
(Pre-pubertal)	3	Thyroid	634	3	Leukemia	237
	4	Colon	220	4	Bladder	209
	5	Bladder	212	5	Thyroid	115
	6	Leukemia	185	6	Prostate	93
	7	Ovary	104	7	Stomach	76
	8	Stomach	101	8	Liver	61
	9	Uterus	50			
	10	Liver	28			
		All Cancers	4777		All Cancers	2563
15 years	1	Breast	553	1	Colon	204
(pubertal)	2	Lung	417	2	Lung	180
	3	Thyroid	178	3	Bladder	127
	4	Colon	134	4	Leukemia	105
	5	Bladder	129	5	Prostate	57
	6	Leukemia	76	6	Stomach	40
	7	Stomach	61	7	Liver	36
	8	Ovary	60	8	Thyroid	33
	9	Uterus	30			
	10	Liver	16			
		All Cancers	2064		All Cancers	1182
60 years	1	Lung	201	1	Colon	94
(Menopausal)	2	Bladder	64	2	Leukemia	82
	3	Colon	62	3	Lung	81
	4	Leukemia	57	4	Bladder	66
	5	Breast	31	5	Prostate	26
	6	Stomach	27	6	Stomach	20
	7	Ovary	18	7	Liver	14
	8	Uterus	9	8	Thyroid	0.3
	9	Liver	7			
	10	Thyroid	1			
		All cancers	586		All cancers	489

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
