# Peer review of "Modulation of Secondary Cancer Risks from Radiation Exposure by Sex, Age and Gonadal Hormone Status: Progress, Opportunities and Challenges"

_jpm, 2022, doi:10.3390/jpm12050725_

Round 1
Reviewer 1 Report
A review article should be well balanced an should carry high degree of fidality to the literature. Second cancer from radiation is relatively rare and teasing the result requires a high level of statistics. Authors failed to refer classical work by Werner, Modan, Ron, Boice and Stovall. Even more contemporary publication such Kry et al should have been used.
Authors have emphasized too much on TRT which only compromised <1% of the entire radiation management. Such data may be emerging and does not justify a major portion of the article.
A few suggested publications are listed below:
- Werner A, Modan B, Davidoff D. Doses to brain, skull and thyroid, following x-ray therapy for tinea capitis. Physics in Medicine and Biology. 1968;13(2):247-58.
- Modan B, Ron E, Werner A. Thyroid cancer following scalp irradiation. Radiology. 1977;123(June):741-4.
- Ron E, Modan B. Thyroid and other neoplasms following childhood scalp irradiation. In: Boice JD, Fraumeni JF, editors. Radiation Carcinogenesis: Epidemiology and Biological Significance. 26. New York: Raven Press; 1984. p. 139-51.
- Modan B, Chetrit A, Alfandary E, Katz L. Increased risk of breast cancer after low-dose irradiation. Lancet. 1989;1:629-31.
- Shore RE, Woodard E, Hildreth N, Dvoretsky P, Hempelmann L, Pasternack B. Thyroid tumors following thymus irradiation. J Nat Cancer Inst. 1985;74(6):1177-84.
- Kry SF, Followill D, White RA, Stovall M, Kuban DA, Salehpour M. Uncertainty of calculated risk estimates for secondary malignancies after radiotherapy. Int J Radiat Oncol Biol Phys. 2007;68(4):1265-71.
- Stovall M, Blackwell CR, Cundiff J, Novack DH, Palta JR, Wagner LK, et al. Fetal dose from radiotherapy with photon beams: Report of AAPM radiation therapy committee task group No. 36. Medical Physics. 1995;22(1):63-82.
- Stovall M, Donaldson SS, Weathers RE, Robison LL, Mertens AC, Winther JF, et al. Genetic effects of radiotherapy for childhood cancer: gonadal dose reconstruction. Int J Radiat Oncol Biol Phys. 2004;60(2):542-52.
- Stovall M, Smith SA. Tissue doses from radiotherapy of cancer of the uterine cervix. Med Phys. 1989;16(5):726-33.
- Stovall M, Smith SA, Langholz BM, Boice JD, Jr., Shore RE, Andersson M, et al. Dose to the contralateral breast from radiotherapy and risk of second primary breast cancer in the WECARE study. Int J Radiat Oncol Biol Phys. 2008;72(4):1021-30.
- Travis LB, Hill D, Dores GM, Gospodarowicz M, van Leeuwen FE, Holowaty E, et al. Cumulative absolute breast cancer risk for young women treated for Hodgkin lymphoma. J Natl Cancer Inst. 2005;97(19):1428-37.
- Travis LB, Hill DA, Dores GM, Gospodarowicz M, van Leeuwen FE, Holowaty E, et al. Breast cancer following radiotherapy and chemotherapy among young women with Hodgkin disease. Jama. 2003;290(4):465-75.
- Tucker MA, D'Angio GJ, Boice JD, Strong LC, Li FP, Stovall M, et al. For the late effects study group. Bone sarcomas linked to radiotherapy and chemotherapy in children. New England Journal of Medicine. 1987;317:588-93.
Author Response
We thank the reviewer for their thoughtful comments. Our detailed responses to all suggestions and comments can be found below.
- A review article should be well balanced an should carry high degree of fidality to the literature.
Our review, titled “Modulation of Secondary Cancer Risks from Radiation Exposure by Sex, Age and Gonadal Hormone Status: Progress, Opportunities and Challenges” is not meant to cover the field of radiation induced cancer but rather review recent and current literature addressing the contribution of biological sex, age and gonadal hormone status to this phenomenon. Consequently, publications that do not report incidence or risk by sex or gonadal hormone status were not included.
- Second cancer from radiation is relatively rare …
Regarding the magnitude of the problem, as far as the population at large is concerned, second cancer from radiation is indeed relatively rare; but it is growing. As eloquently stated stated in Kumar et al. (2012), “More than half of all cancer patients receive radiotherapy as a part of their treatment. With the increasing number of long-term cancer survivors, there is a growing concern about the risk of radiation induced second malignant neoplasm [SMN]. This risk appears to be highest for survivors of childhood cancers” (Kumar, S. Second malignant neoplasms following radiotherapy. Int. J. Environ. Res. 2012, 9, 4744–4759. DOI:10.3390/ijerph9124744)
Notably, even in this high risk group, the risk is not equally distributed and there are subgroups of the population (most notably, young girls) for whom (Modifiable!) medical irradiation poses substantial risks of death from secondary cancer: “The cumulative incidence of breast cancer by age 50 years was 30% with a 35% incidence among Hodgkin lymphoma survivors. Breast cancer associated mortality at 5 and 10 years was also substantial (12% and 19%, respectively)” (Moskowitz CS, Chou JF, Wolden SL, Bernstein JL, Malhotra J, Novetsky Friedman D, et al. . Breast cancer after chest radiation therapy for childhood cancer. J Clin Oncol 2014; 32: 2217–23.doi: 10.1200/JCO.2013.54.4601) Cited in:
McKeown, S. R., Hatfield, P., Prestwich, R. J., Shaffer, R. E., & Taylor, R. E. (2015). Radiotherapy for benign disease; assessing the risk of radiation-induced cancer following exposure to intermediate dose radiation. The British journal of radiology, 88(1056), 20150405. https://doi.org/10.1259/bjr.20150405
- …and teasing the result requires a high level of statistics.
We absolutely agree with the reviewer that teasing out the effects of sex and hormonal status requires a high level of statistics (we take this to mean studies large enough to power a sex-stratified analysis), which is the reason we have devoted the most attention to the publications about atomic bomb survivors, representing the largest population ever studied in this context. This is also the reason that studies which do not report effects of biological sex (possibly because they lack statistical power to perform such analysis) are not cited in our review.
Authors failed to refer classical work by Werner, Modan, Ron, Boice and Stovall. Even more contemporary publication such Kry et al should have been used. A few suggested publications are listed below:
We thank the reviewer for his helpful suggestion of additional citations, the majority of which have been incorporated into the revised version of the manuscript. When we have not done this, it was for the reasons stated above. NB, in the context of our review, the most recent publications from any cohort, reflecting the longest follow-up, are cited since radiation-induced cancers may take decades to manifest, as noted by Modan et al (1998) and other publications cited in our review. In this spirit, we have added the paper by Boaventura et al. (2014) describing sex differences in thyroid cancer after scalp irradiatio rather than adding earlier papers from the Israeli study which was already cited (Sadtetzky et al. 2006).
Authors have emphasized too much on TRT which only compromised <1% of the entire radiation management. Such data may be emerging and does not justify a major portion of the article.
Since the special issue for which this review was written, titled ” Personalizing medicine by sex, gender and hormonal status: progress , opportunities and challenges” emphasized progress, opportunities and challenges, ie recent development and future perspectives rather than historical perspective, we chose to emphasize TRT, which is the fastest moving field in radiation therapy and therefore the filed with the fastest rate of progress as well as opportunities and challenges.
Reviewer 2 Report
This is a useful review of age and sex dependencies on cancer risk. The only concerns I see are:
1) the most recent large series of tissue specific papers from ~2017-2020 on analysis of the atomic bomb survivor cancer risks are not cited. These are papers by Brenner et al on breast cancer risks, Richardson et al etc largely published in the Radiation Research journal. The Brenner paper has very detailed models on role of menarche on breast cancer risk from radiation.
2) the role of longevity and race/ethnic group are not really covered. These factors impact age and sex dependencies of radiation risks. For example Cucinotta and Saganti in Scientific Reports (2022).
3) the influence of tobacco products on lung and a few other cancer types is barely discussed and influences radiation risks.
4) the biological basis for differences in men and women on radiation lung cancer risk is not known and should be mentioned in the conclusions.
Author Response
Reviewer 2
We thank the reviewer for their thoughtful comments. Our detailed responses to all suggestions and comments can be found below.
This is a useful review of age and sex dependencies on cancer risk.
Thank you!
The only concerns I see are:
- the most recent large series of tissue specific papers from ~2017-2020 on analysis of the atomic bomb survivor cancer risks are not cited. These are papers by Brenner et al on breast cancer risks, Richardson et al etc largely published in the Radiation Research journal. The Brenner paper has very detailed models on role of menarche on breast cancer risk from radiation.
Thank you for this helpful suggestion. The Brenner paper on menarche has been added to the revised manuscript, as well as a few other recent papers about effects of pregnancy and lactation on cancers secondary to accidental exposures (Cahoon et al., 2021, Rivkind et al., 2020)
- the role of longevity and race/ethnic group are not really covered. These factors impact age and sex dependencies of radiation risks. For example Cucinotta and Saganti in Scientific Reports (2022).
We thank you for this suggestion. The suggested paper has been added to the reference list, although the results do not support an interaction between sex and race/ethnicity-rather,
Effects of race and ethnicity are miniscule relative to effects of sex
- the influence of tobacco products on lung and a few other cancer types is barely discussed and influences radiation risks.
We added a citation to the recent atomic bomb survivors paper which addresses smoking, (Furukawa, K., Preston, D. L., Lönn, S., Funamoto, S., Yonehara, S., Matsuo, T., Egawa, H., Tokuoka, S., Ozasa, K., Kasagi, F., Kodama, K., & Mabuchi, K. (2010). Radiation and smoking effects on lung cancer incidence among atomic bomb survivors. Radiation research, 174(1), 72–82. https://doi.org/10.1667/RR2083.1), which did not show a differential effect/interaction with sex (ie the size of the sex difference was the same in the whole population and in non-smokers). NB, as mentioned in our response to reviewer 1 above, ours is not a review on radiation risks in general but rather a focused review on the contribution of sex and gonadal hormones to these risks. The influence of other factors is relevant if there is evidence that they interact with sex and influence the size of the sex difference.
4) the biological basis for differences in men and women on radiation lung cancer risk is not known and should be mentioned in the conclusions.
Thank you for this comment.
As suggested, we have added a sentence to the conclusions to reflect this fact.
“Smaller cohort studies in humans and animals also show large changes in cell proliferation rates, radiotracer accumulation and target density in female reproductive and non-reproductive organs, including breast, thyroid and brain, in conjunction with physiological changes in gonadal hormone (especially estradiol) levels such as occur during the menstrual cycle and menopause. NB, while these effects may explain higher propensity of females to radiation induced cancers of the breast and the thyroid, they do not explain the sex difference in lung cancer, the biological origin of which is still unknown.”
Round 2
Reviewer 1 Report
Grammar and spaces in revised manuscript is visible. Please proof read and correct. Capital cases in the middle of sentence is annoying.